# Effects of cannabis ingestion on endometriosis-associated pelvic pain and related symptoms

**Justin Sinclair**[1]☺*, **Laura Collett**[2]☺, **Jason Abbott**[3‡], **David W. Pate**[1‡], **Jerome Sarris**[1,4‡], **Mike Armour**[1,5‡]

**1** NICM Health Research Institute, Western Sydney University, Sydney, NSW, Australia, **2** Bristol Trials Centre, University of Bristol, Bristol, United Kingdom, **3** School of Women's and Children's Health, UNSW, Sydney, NSW, Australia, **4** Professorial Unit, The Melbourne Clinic, Department of Psychiatry, University of Melbourne, Melbourne, VIC, Australia, **5** Translational Health Research Institute, Western Sydney University, Sydney, NSW, Australia

☺ These authors contributed equally to this work.
‡ These authors also contributed equally to this work.
* 19948081@student.westernsydney.edu.au

## Abstract

### Background

The use of cannabis for symptoms of endometriosis was investigated utilising retrospective archival data from Strainprint Technologies Ltd., a Canadian data technology company with a mobile phone application that tracks a range of data including dose, mode of administration, chemovar and their effects on various self-reported outcomes, including pelvic pain.

### Methods

A retrospective, electronic record-based cohort study of Strainprint™ users with self-reported endometriosis was conducted. Self-rated cannabis efficacy, defined as a function of initial and final symptom ratings, was investigated across the included symptom clusters of cramps, pelvic pain, gastrointestinal pain, nausea, depression, and low libido. Cannabis dosage form, dose and cannabinoid ratio information was also recorded.

### Results

A total number of 252 participants identifying as suffering endometriosis recorded 16193 sessions using cannabis between April 2017 and February 2020. The most common method of ingestion was inhalation (n = 10914, 67.4%), with pain as the most common reported symptom being treated by cannabis (n = 9281, 57.3%). Gastrointestinal symptoms, though a less common reason for cannabis usage (15.2%), had the greatest self-reported improvement after use. Inhaled forms had higher efficacy for pain, while oral forms were superior for mood and gastrointestinal symptoms. Dosage varied across ingestion methods, with a median dose of 9 inhalations (IQR 5 to 11) for inhaled dosage forms and 1 mg/mL (IQR 0.5 to 2) for other ingested dosage forms. The ratio of THC to CBD had a statistically

**Data Availability Statement:** The data underlying this manuscript was based on data collected by Strainprint Technologies Ltd mHealth app between

April 2017 until February 2020. Permission to use this data was granted by David Berg, President of Strainprint Technologies Ltd. To request access to this data please contact David Berg at dave@strainprint.ca. The authors confirm they did not have any special access privileges that others would not have.

**Funding:** The author(s) received no specific funding for this work.

**Competing interests:** I have read the journal's policy and the authors of this manuscript have the following competing interests: As a medical research institute, NICM Health Research Institute receives research grants and donations from foundations, universities, government agencies, individuals and industry. Sponsors and donors also provide untied funding for work to advance the vision and mission of the Institute. The authors declare no competing financial interests. JS is the recipient of a Western Sydney University Postgraduate Research Scholarship. J.Sarris has received either presentation honoraria, travel support, clinical trial grants, book royalties, or independent consultancy payments from: Australian Natural Therapeutics Group (being a scientific advisor to Australian Natural Therapeutics Group: a cannabis grower and manufacturer), Integria Healthcare & MediHerb, Pfizer, Scius Health, Key Pharmaceuticals, Taki Mai, Fiji Kava, FIT-BioCeuticals, Blackmores, Soho-Flordis, Healthworld, HealthEd, HealthMasters, Kantar Consulting, Angelini Pharmaceuticals, Grunbiotics, Polistudium, Research Reviews, Elsevier, Chaminade University, International Society for Affective Disorders, Complementary Medicines Australia, SPRIM, Terry White Chemists, ANS, Society for Medicinal Plant and Natural Product Research, Sanofi-Aventis, Omega-3 Centre, the National Health and Medical Research Council, CR Roper Fellowship. This does not alter our adherence to PLOS ONE policies on sharing data and materials.

significant, yet clinically small, differential effect on efficacy, depending on method of ingestion.

## Conclusions

Cannabis appears to be effective for pelvic pain, gastrointestinal issues and mood, with effectiveness differing based on method of ingestion. The greater propensity for use of an inhaled dosage delivery may be due to the rapid onset of pain-relieving effects versus the slower onset of oral products. Oral forms appeared to be superior compared to inhaled forms in the less commonly reported mood or gastrointestinal categories. Clinical trials investigating the tolerability and effectiveness of cannabis for endometriosis pain and associated symptoms are urgently required.

## Introduction

Endometriosis is a common, chronic inflammatory condition in women characterised by the presence of endometrial-like tissue found outside the uterus [1]. Prevalence rates of the disease have been estimated at between 5% [2] and 11% [3] of reproductive-aged women, impacting an estimated 176 million women worldwide. [4]. In addition, endometriosis can also affect post-menopausal women, ranging from between 2–5% of cases [5]. Endometriosis is associated with a variety of symptoms including chronic pelvic pain, fatigue, dysmenorrhoea (period pain), dyspareunia (painful sex), dyschezia (painful bowel movements) and dysuria (pain related to urination) [6,7]. In addition, co-morbid anxiety and/or depression in the endometriosis cohort is frequently reported [8,9], along with irritable bowel syndrome (IBS)-like gastrointestinal symptoms [10,11] and significant fatigue [6]. These symptoms contribute to substantial reductions in many aspects of quality-of-life including social, academic, work and sexual relationships [12,13]. In addition, endometriosis causes a noteworthy cost of illness burden to the economy, mostly due to lost productivity [14–16].

Recent studies have suggested that a dysfunction in the endocannabinoid system (ECS) is present in endometriosis patients [17,18], and that aspects of endometriosis-associated pain may be targeted by modulating the ECS [19].

Previous research on the use of illicit cannabis in women with endometriosis has shown promise in the treatment of endometriosis pain and co-morbid symptoms such as poor sleep, gastrointestinal upset and mood disorders [20–22]. However, these findings were all based on self-reported data and could not be reliably used to determine the relative effectiveness of different modes of cannabis administration, dose amount, cannabinoid ratios, or other determining factors (such as patient age), all of which may influence clinical outcomes.

This study sought to investigate the self-rated effectiveness, dosage forms, dosage amount and cannabinoid ratios of quality-assured legal cannabis products that women are using in the Canadian regulated market, via their use of the Strainprint™ smartphone application ("app"), which is used to track legal cannabis product usage for medicinal purposes.

## Materials & methods

### Procedure

Strainprint Technologies Ltd. is a Canadian-based data and analytics company that provides a free mobile phone app that allows individuals to prospectively track medicinal cannabis usage,

including the varieties of cannabis, dosage form, dosage and changes in symptom severity of users over time. During the initial session, users enter basic demographic information such as age and gender, and then identify the medical conditions or symptoms they are experiencing as part of the onboarding process. Prior to using cannabis, users open the app and identify the symptom for which they will be using cannabis, which is then recorded as a "session". A session was defined as the process by which participants select the symptom(s) they are currently experiencing (allowing categorization), self-rate the severity of the symptom(s) using an 11-point numerical rating scale (0 being lowest severity to 10 being highest) prior to cannabis use, select the product and dosage form they will use to address the aforementioned symptom (s), and record the dose taken. Product choices are pre-populated from a list of over 6500 laboratory-verified products in the Strainprint™ database offered by licenced medicinal *Cannabis* cultivators and distributors, including the records of the *delta*-9-tetrahydrocannabinol (THC) and cannabidiol (CBD) concentrations and ratios, in addition to dominant terpene expressions within the database. Whilst the baseline severity of each symptom is obtained prior to taking the dose, the app prompts users after a set time period (20 minutes for inhaled forms and 90 minutes for oral forms) to revaluate the specific symptom severity scores.

Archival data in Excel format was provided by Strainprint Technologies Ltd. for the period April 2017 until February 2020. During this period, only pre-populated and lab-verified product selection was available, with no ability for manual entry by users. Demographic data and the breakdown of clinical indications and dosage forms from the data set are noted in Table 1. To facilitate multilevel analysis, symptoms were classified into clinically meaningful subgroups *a priori*, arranged as either "Pain" (including pelvic pain and cramps), "Gastrointestinal" (including gastrointestinal pain and nausea) and "Mood" (incorporating depression and low libido). Further characterisation was applied to the dosage form utilised by the users and included "Oral" (including edibles [mg], oils [mL], capsules [mg], tinctures [mL] or sprays [mL], "Topical" (including transdermal and topical) and "Inhaled" forms ("puffs" for smoked and vapourised).

Due to the nature of the research project utilising fully anonymised retrospective archival datasets, this project was exempt from ethical review by the Western Sydney University Human Ethics Committee as per the National Health & Medical Research Council (NHMRC) National Statement [23]. All individuals who register to use the Strainprint™ app signed a Consent to Collection and Use of Data form for research purposes

## Data analysis

Self-rated cannabis efficacy, defined as a function of initial and final symptom ratings before and after cannabis use was utilised.

$$\text{Efficacy} = \frac{((x - y)/x) + ((x - y)/10)}{2}$$

Where $x$ = initial symptom rating (score out of 10) and $y$ = final symptom rating (score out of 10).

Statistical analysis was carried out using multilevel linear regression to examine the effect of various independent factors on efficacy rating. Participants and sessions were classed as random effects to take into account dependence between observations in the same session from the same user, and all other factors were classed as fixed effects. Additionally, the ratio of THC to CBD was hypothesised to be an important determinant of efficacy, and due to the fact that some participants' recorded cannabinoids with zero amounts of THC or CBD, trace levels

**Table 1. Demographic of Strainprint<sup>TM</sup> users with endometriosis.**

| | |
|---|---|
| Number of Strainprint<sup>TM</sup> users | 252 |
| Total number of recorded cannabis sessions | 16187 |
| | **Mean (SD)** |
| Average age at first use of Strainprint<sup>TM</sup> app | 33.10 (7.8) |
| Average months using Strainprint<sup>TM</sup> app per user | 5.5 (7.4) |
| Average number of recorded cannabis sessions per user | 64.2 (170.8) |
| Average number of clinical indications per user | 2.6 (1.5) |
| Average number of dosage forms per user | 2.0 (1.4) |
| **Dosage form breakdown (n = 16187 sessions)** | **N (%)** |
| *Inhaled products* | |
| Vaporised | 6575 (40.6) |
| Smoked | 4191 (25.9) |
| Concentrate | 132 (0.84) |
| Dab bubbler/rig | 16 (0.10) |
| *Orally ingested products* | |
| Oil | 4041 (25.0) |
| Pill | 558 (3.5) |
| Edible | 389 (2.4) |
| Spray | 90 (0.56) |
| Tincture | 90 (0.56) |
| Oral | 52 (0.32) |
| *Topical products* | |
| Topical | 33 (0.20) |
| Transdermal | 20 (0.12) |
| **Clinical indications breakdown (n = 16192 sessions)** | **N (%)** |
| Pelvic Pain | 6864 (42.4) |
| Gastrointestinal Distress | 2461 (15.2) |
| Cramps | 2417 (14.9) |
| Nausea | 2254 (13.9) |
| Depression | 2138 (13.2) |
| Low Libido | 53 (0.33) |

(0.0001) of THC or CBD were assigned to these data points for the purpose of modelling, so that ratios could be calculated and missing data minimised.

## Results

A total number of 252 participants were identified as having endometriosis, with a total of 16,193 sessions. Six sessions were excluded from this dataset due to ingestion method (suppository, n = 1), outlying THC ($>$500 mg/mL, n = 3), outlying CBD ($>$500 mg/mL, n = 1), and negative dosage (n = 1), yielding a total of 16,187 sessions included in the final analysis. Prevalence of cannabis use was rated highest for the pain subgroup (n = 9281/57.3%), followed by gastrointestinal (n = 4715/29.1%) and mood (n = 2191/13.5%).

The mean age at the beginning of each session of women identifying as having endometriosis within the Strainprint<sup>TM</sup> dataset was 35 years, with the majority using the app for approximately 5.5 months (See Table 1). Pulmonary administration was the favoured dosage form, with inhaled forms accounting for 67.4% of all dosage forms utilised by the cohort: Vapouriser (40.6%), Smoked (25.9%), dab bubbler or concentrate (0.9%). Orally ingested dosage forms

were the next most utilised, accounting for 32.3% of the cohort: Oils (25.0%), Pills (3.5%), Edible products (2.4%), Tincture as spray or oral drops (1.4%). Topicals (0.2%) and transdermals (0.1%) had minimal reported usage in this dataset.

Pelvic Pain (42.4% of all participants) was the primary clinical indication for use of cannabis, with Gastrointestinal Distress (15.2%) and Cramps (14.9%) following in frequency. Nausea (13.9%) and Depression (13.2%) were also notable findings, with Low Libido (0.3%) being the least reported symptom.

Dosage varied according to ingestion method, but not for the symptom being treated, with the highest median dose being taken by those using inhalation (9 mg/mL or puff, IQR: 5, 11, n = 10,914). Lowest median dose was taken by those treating pain and mood via oral dosage forms, with a median dose of 1 mg/mL, capsule or piece (with an IQR of 0.5 to 2, n = 5220). The median dose for topical application was 2 mg/mL (IQR: 1.5 to 20, n = 53).

THC and CBD levels (and THC/CBD ratios) varied widely, depending on the ingestion method (Fig 1). Inhaled methods typically had a high THC to CBD ratio, with a median % or mg/mL THC of 16 (IQR 8 to 19.6), a median % or mg/mL CBD of 0.07 (IQR 0 to 8.01) and a median THC/CBD ratio of 90 (IQR 0.6 to 304). Ingested dosage forms exhibited the opposite tendency, having a high CBD to THC ratio, with a median for THC of 2.5 (IQR 1.0 to 12), a median for CBD of 13.5 (IQR 5 to 25) and a median THC/CBD ratio of 0.08 (IQR 0.04 to 1). The median amounts of THC and CBD for topical dosage forms was equal (median ratio of 1, IQR 0.4 to 13.2). The THC/CBD ratio remained similar as median ratios across the symptoms:

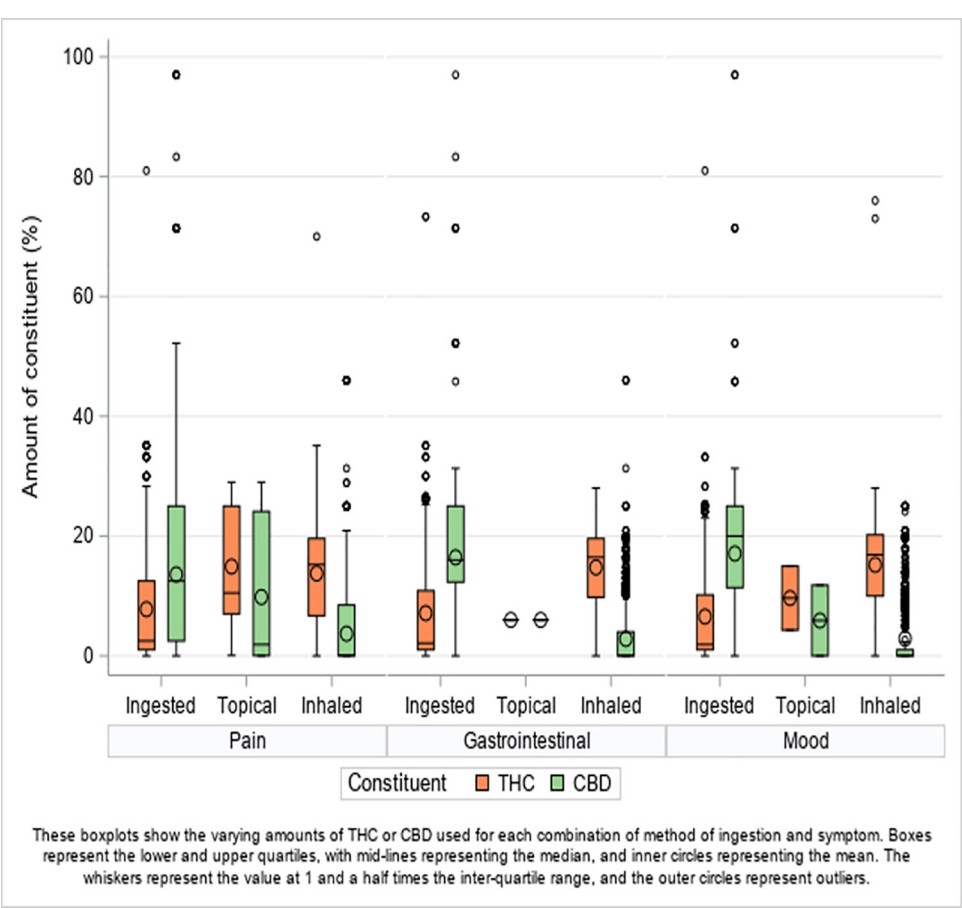

These boxplots show the varying amounts of THC or CBD used for each combination of method of ingestion and symptom. Boxes represent the lower and upper quartiles, with mid-lines representing the median, and inner circles representing the mean. The whiskers represent the value at 1 and a half times the inter-quartile range, and the outer circles represent outliers.

**Fig 1. THC and CBD by symptom group and ingestion method.**

**Table 2. Final model results.**

| Factor | Factor level | Estimate (95% CI) | P-value |
|---|---|---|---|
| Intercept | | 31.98 (31.26, 32.71) | <0.0001 |
| Centred age (age at session–mean age) | | 0.26 (0.20, 0.31) | <0.0001 |
| $Log_{10}$(THC/CBD) | | 1.72 (1.22, 2.23) | <0.0001 |
| Symptom (class) | Pain | Reference category | |
| | Gastrointestinal | 9.02 (8.15, 9.90) | <0.0001 |
| | Mood | 0.66 (-0.58, 1.90) | 0.2952 |
| Ingestion method (class) | Inhaled | Reference category | |
| | Ingested | -2.11 (-3.23, -1.00) | 0.0002 |
| | Topical | 13.64 (6.11, 21.17) | 0.0004 |
| Interaction: $Log_{10}$(THC/CBD) and ingestion method (class) | Inhaled | | |
| | Ingested | -4.75 (-5.62, -3.88) | <0.0001 |
| | Topical | -2.15 (-8.51, 4.22) | 0.5084 |
| Interaction: Centred age (age at session–mean age) and dosage | | -0.007 (-0.009, -0.004) | <0.0001 |
| Interaction: $Log_{10}$(THC/CBD) and dosage | | 0.03 (0.008, 0.04) | 0.0039 |
| Interaction: Symptom (class) and Ingestion method (class) | Gastrointestinal * Inhaled | Reference category | |
| | Mood * Inhaled | Reference category | |
| | Pain * Ingested | Reference category | |
| | Pain * Topical | Reference category | |
| | Pain * Inhaled | Reference category | |
| | Gastrointestinal * Ingested | 7.82 (6.26, 9.37) | <0.0001 |
| | Gastrointestinal * Topical | -3.04 (-40.64, 34.57) | 0.8743 |
| | Mood * Ingested | 5.16 (2.96, 7.36) | <0.0001 |
| | Mood * Topical | 33.90 (1.34, 66.46) | 0.0413 |

Pain 0.95 (IQR 0.1 to 197) *vs.* Gastrointestinal Distress 0.88 (IQR 0.08 to 242) *vs.* Mood 0.89 (IQR 0.08 to 220), although approximately 8% less THC than CBD was used overall.

Results of the multilevel model are reported in Table 2. Overall, cannabis was shown to have a positive effect, with a modelled mean baseline efficacy rating of 31.98 (95% CI 31.26 to 32.71, p<0.0001). The symptom with the largest effect was seen for gastrointestinal symptoms, with an estimated increase in efficacy of 9.02 (95% CI 8.15 to 9.90, p<0.0001) compared to pain. Mood symptoms had effects not significantly different from baseline of pain. Despite there being relatively few participants reporting use of topical administrations, this had the strongest effect of all the ingestion methods, with an estimated increase in efficacy of 13.64 (95% CI 6.11 to 21.17, p = 0.0004) in comparison to the baseline for inhalation. Orally ingested products had a slightly lower efficacy than the baseline observed for inhaled cannabis (-2.11 with 95% CI –3.23 to –1.00, p = 0.0002). Ingestion methods varied according to symptom. Examination of the interaction between symptom group and ingestion method revealed an increase in efficacy when taking oral cannabis for gastrointestinal symptoms, with an estimate of 7.82 (95% CI 6.26 to 9.37, p<0.0001) adding to the effect of oral and gastrointestinal seen alone. There was also an increase in efficacy for those treating mood with orally ingested forms (5.16 with 95% CI 2.96 to 7.36, p<0.0001). Effects of topical administration on gastrointestinal and mood symptoms were intriguing, but should be treated with caution as these subgroups had minimal usage in this dataset.

The effect of THC/CBD ratio gave an increase in efficacy of 1.72 (95% CI 1.22 to 2.23, p<0.0001) for every ten-fold increase in THC. Age also had a statistically significant effect,

with an increase in efficacy of 0.26 (95% CI 0.20 to 0.31, p<0.0001) for each year increase in age, with older women having better overall efficacy.

As seen in Fig 1, the relationship between THC and CBD levels differed between ingestion methods, and the model showed that efficacy also differed subsequently. There was no evidence of a difference in efficacy according to the ratio of THC to CBD between topical and inhaled treatments. However, there was a difference between inhaled and orally ingested cannabis in combination with the ratio of THC to CBD, where for ingested methods, there was a reduction in efficacy of 4.75 (95% CI 5.62 to 3.88) for every 10-fold increase in THC compared to CBD (an increase in efficacy for every 10-fold increase in CBD compared to THC).

## Discussion

Our findings demonstrate that women self-reporting endometriosis used cannabis to manage a variety of their symptoms, including pelvic pain and gastrointestinal tract (GIT) symptoms. Inhaled forms of cannabis were the most commonly used, followed by oral ingestion, where both were similarly efficacious in terms of alleviating symptoms. In particular, there was no difference between these two routes of administration with respect to pelvic pain. This may be due to the different time points that effectiveness scores were taken in the Strainprint$^{TM}$ app: 20 minutes from intake for inhaled *vs*. 90 minutes for oral. Therefore, the effectiveness scores are unlikely to capture this important difference in speed of onset for pain relief. Orally ingested forms appeared to be superior compared to inhaled forms for the less commonly reported mood or gastrointestinal issues.

Age-associated increases in effectiveness was also observed within the dataset, with better subject-perceived effectiveness of cannabis preparations existing for a variety of symptoms, whether using inhaled or orally ingested dosage forms, seeming to increase with greater age. Why perception of effectiveness increased with age is unknown, but could plausibly be due to age-related change to organ function causing increased sensitivity to drugs/medications [24]. changing hormone levels [25] or the possible decline in endocannabinoid system functioning associated with ageing [26,27]. It may also be due to the changes in pain perception that occur over time in people with endometriosis [28,29].

The preferred dosage form in previous research investigating illicit cannabis usage in endometriosis cohorts in Australia and New Zealand was via the inhaled route (including smoked and vapourised), with 61.9% of Australian [20] and 67.8% of New Zealand [21] respondents favouring this method of administration. Whilst it is plausible that inhaled forms are favoured within illicit markets due to a scarcity of other dosage forms, such a similarity within our legal cannabis dataset suggests other factors may be involved in the choice to use inhaled forms.

A key factor in the use of inhaled forms for endometriosis may be that there is a faster onset of pharmacological action, and therefore symptom reduction, by cannabinoids (and terpenes) administered via the inhaled route, usually within 5–10 minutes, in contrast to the 45–180 minutes for oral dosage forms [30]. The absorption and bioavailability of THC and other cannabinoids via smoking has been typically reported between 2% and 56% due to variables such as the cannabis incineration temperature, the inhalation number and duration, and inhalation volume/hold-time, *etc*. However, it is generally higher than the slow rate of absorption and poor bioavailability (4% to 20%) of oral dosage forms [31]. The difference in this speed of onset means that both routes of administration may play a vital role as women with endometriosis have both chronic pain and more acute pain, often described as endometriosis "flares" [32,33], which are characterised as episodes of acute and intense breakthrough pain. Such sudden-onset pain events require a fast analgesic onset for immediate symptom relief and may explain the prevalence of the inhaled route for dose delivery. This may also be relevant for

patients with Deep Infiltrating Endometriosis (DIE) who suffer from pelvic floor muscle hypertonic dysfunction [34], whereby the muscle relaxant activity associated with cannabis [35] may also play an important role. Finally, another factor favouring the inhaled route is that women have previously reported that they have chosen inhaled forms due its ease of dose control, in order to titrate a balance between pain relief and impairment [36]. Lastly, women may be favouring inhaled dosage forms due to familiarity from previous recreational use [20,21], with smoked cannabis being the most common method of administration historically [37].

The effect of changing the THC:CBD ratio appeared to have only a minor impact on pelvic pain, suggesting that a precise titration of THC is important for symptom management, but that higher levels may overly impair or increase known side-effects, potentially causing issues with participants being able to perform Activities of Daily Living (ADL). This may also suggest that women who use illicit cannabis for therapeutic purposes may not always receive the ideal cannabinoid ratio for pelvic pain as an oral dosage form due to unknown cannabinoid levels and lack of standardisation. Average illicit cannabis THC potency has increased in strength over the last 50 years due to the decentralised development of underground selective-breeding programs and widespread adoption of the "sinsemilla" (*i.e.*, without seeds) growing technique, combined with a consequent market erosion for lower potency seeded cannabis. Although connoisseur high-potency varieties were available historically, cannabis in the 1960s more commonly had a typical THC concentration of between 1–5%. This compares to chemovars today commonly ranging in content between 15–30% THC [37,38], usually at the expense of CBD content. This higher ratio of THC in illicit cannabis may have contributed to the pain reduction found in previous research [20,21] featuring an endometriosis cohort using illicit cannabis. However, the implied high absolute levels of THC may not be needed to gain optimal symptom control, as lower doses are readily implemented via inhaled dose titration. Nevertheless, future research is needed to explore this aspect of dosing in more detail.

Plausible mechanisms exist for improvements in the GIT and mental health symptoms reported. Although THC plays a role in gastrointestinal symptoms, including nausea and vomiting, CBD has documented anti-inflammatory and antioxidant activity [39], with a noted fatty acid amide hydrolase (FAAH) inhibition demonstrated to induce anti-inflammatory effects in the GIT [40,41]. CBD has documented anti-emetic and anti-nausea activity [42], and emergent evidence of CBD being of benefit in gastrointestinal conditions has been reported [43,44]. Further, a clinical endocannabinoid deficiency [45,46] has been posited for conditions such as irritable bowel syndrome, a common co-morbid diagnosis in people with endometriosis [47,48]. However, CBD is not a ligand for the CB1 and CB2 receptor, so other mechanisms must be extant. For example, CBD exhibits known anxiolytic and antidepressant activity via modulation of the serotonin (5HT1A) receptor [49,50].

Absorption of CBD and other cannabinoids may be increased with oral dosage forms by utilising an oil (*e.g.*, olive oil, medium-chain triglycerides, etc) as the carrier/excipient, or if taken with meals that contain fats, resulting in a longer duration of effect (6–8 hours) [30]. Oral administration also provides patients with more stable plasma levels and, therefore, a more sustained therapeutic benefit. This route also provides higher concentrations of CBD compared to inhaled products, which may explain why there was a greater reported effectiveness in mood and GIT symptoms in the present data set.

Whilst topical dosage forms demonstrated a notable effect on pain compared to inhalation, the actual number of participants reporting use of this dosage form was very small, therefore caution should be applied in interpreting or extrapolating from this data. Topical applications offer a novel method of dose delivery and may be useful for localised symptoms and effects, but do have noted variability in both onset and duration of systemic effect [30]. With a dearth of evidence for this dosage form, both generally and in the endometriosis cohort particularly,

further research is required to learn about cannabinoid tolerability, ratios and extent of therapeutic effect.

Limitations of this study include that the data was cross-sectional and that it captured regular users rather than *de novo* use longitudinally. Additionally, during the timeline of this captured dataset (April 2017—February 2020), the predominant dosage form reported by the Strainprint<sup>TM</sup> app for the Canadian regulated medicinal cannabis market was heavily weighted towards dried cannabis flos (flower). This suggests that the higher percentage of cannabis-inhaling users presented in the data may have reflected this predominance and is not necessarily due only to perceived effectiveness.

Conversely, given the average use of the Strainprint<sup>TM</sup> app was approximately 6 months, coupled with an average of 64.2 recorded medication sessions per user, a greater proportion of those who experienced improvement may have continued to log data, due to the perceived effectiveness of the app and the cannabis dosage form being utilised.

## Conclusion

With emerging evidence internationally demonstrating that women are utilising illicit cannabis as a self-management strategy for the pain and the associated symptoms of endometriosis, this paper demonstrates that Canadian women are also utilising legally obtained and quality-assured products to manage endometriosis symptoms across domains such as pelvic pain, gastrointestinal symptoms and mood. Cannabis appears to be effective across all reported symptoms, with a noted propensity for inhaled delivery due to the potential increased speed of onset of effects *versus* the slower onset of oral products, particularly for pelvic pain. Conversely, oral forms appeared to be superior for the less reported mood and gastrointestinal categories, possibly due to higher CBD concentrations in the products utilised. Whilst topical products demonstrated a good effect on pain, due to a very small data set, caution should be exercised in interpreting or extrapolating from this data.

The importance of quality-assured and standardised (i.e., cannabinoid potency and ratios) cannabis products to obtain reproduceable clinical results, and to mitigate possible adverse effects due to potential adulteration or contamination of products, is an important clinical consideration for medicinal cannabis moving forward, particularly when considering the urgent need for clinical trials investigating the safety, tolerability and effectiveness of cannabis for endometriosis pain and associated symptoms.

## Acknowledgments

The authors wish to thank Strainprint Technologies Ltd. for their support and data access.

## Author Contributions

**Conceptualization:** Justin Sinclair, Mike Armour.

**Data curation:** Justin Sinclair, Laura Collett.

**Formal analysis:** Laura Collett.

**Investigation:** Justin Sinclair.

**Methodology:** Justin Sinclair, Laura Collett.

**Supervision:** Jason Abbott, David W. Pate, Jerome Sarris, Mike Armour.

**Writing – original draft:** Justin Sinclair, Laura Collett, Mike Armour.

**Writing – review & editing:** Justin Sinclair, Laura Collett, Jason Abbott, David W. Pate, Jerome Sarris, Mike Armour.

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
