## [Decision Letter · Decision Letter 0]

23 Sep 2021

PONE-D-21-19050

Effects of cannabis ingestion on endometriosis-associated pelvic pain and related symptoms.

PLOS ONE

Dear Dr. Sinclair,

Thank you for submitting your manuscript to PLOS ONE. After careful consideration, we feel that it has merit but does not fully meet PLOS ONE’s publication criteria as it currently stands. Therefore, we invite you to submit a revised version of the manuscript that addresses the points raised during the review process.

Please revise the manuscript according to Reviewers’ minor comments.

We look forward to receiving your revised manuscript.

Kind regards,

Diego Raimondo

Academic Editor

PLOS ONE

“The authors wish to thank Strainprint Technologies Ltd. for their support and data access. J. Sarris is supported by an NHMRC Clinical Research Fellowship (APP1125000).”

“I have read the journal's policy and the authors of this manuscript have the following competing interests: As a medical research institute, NICM Health Research Institute receives research grants and donations from foundations, universities, government agencies, individuals and industry. Sponsors and donors also provide untied funding for work to advance the vision and mission of the Institute. The authors declare no competing financial interests. JS is the recipient of a Western Sydney University Postgraduate Research Scholarship. J.Sarris has received either presentation honoraria, travel support, clinical trial grants, book royalties, or independent consultancy payments from: Australian Natural Therapeutics Group (being a scientific advisor to Australian Natural Therapeutics Group: a cannabis grower and manufacturer), Integria Healthcare & MediHerb, Pfizer, Scius Health, Key Pharmaceuticals, Taki Mai, Fiji Kava, FIT-BioCeuticals, Blackmores, Soho-Flordis, Healthworld, HealthEd, HealthMasters, Kantar Consulting, Angelini Pharmaceuticals, Grunbiotics, Polistudium, Research Reviews, Elsevier, Chaminade University, International Society for Affective Disorders, Complementary Medicines Australia, SPRIM, Terry White Chemists, ANS, Society for Medicinal Plant and Natural Product Research, Sanofi-Aventis, Omega-3 Centre, the National Health and Medical Research Council, CR Roper Fellowship.”

Additional Editor Comments (if provided):

Comments from Senior Staff Editor:

We note that one or more reviewers has recommended that you cite specific previously published works. As always, we recommend that you please review and evaluate the requested works to determine whether they are relevant and should be cited. It is not a requirement to cite these works. We appreciate your attention to this request.

Reviewers' comments:

Reviewer's Responses to Questions

**Comments to the Author**

1. Is the manuscript technically sound, and do the data support the conclusions?

Reviewer #1: Yes

2. Has the statistical analysis been performed appropriately and rigorously? 

Reviewer #1: Yes

3. Have the authors made all data underlying the findings in their manuscript fully available?

Reviewer #1: Yes

4. Is the manuscript presented in an intelligible fashion and written in standard English?

Reviewer #1: Yes

5. Review Comments to the Author

Reviewer #1: The study is well thought out and properly written. In the introduction you should add that endometriosis not only affects women of childbearing age, but also postmenopausal (cites "Retroperitoneal endometriosis in postmenopausal woman causing deep vein thrombosis: case report and review of the literature, Ianieri MM, Clin Exp Obstet Gynecol. 2017; 44 (1): 148-150)

The materials and methods are clear as well as the results. The discussion should also be improved by talking about the potential hypertonus pelvic muscles of patients with endometriosis (cite DOI 10.1002 / uog.18924, Mabrouk M 2018) and the potential effect of cannabis at therapeutic dosage.

6. PLOS authors have the option to publish the peer review history of their article (what does this mean?). If published, this will include your full peer review and any attached files.

Reviewer #1: No

---

## [Author Response · Author response to Decision Letter 0]

7 Oct 2021

Please see Response to Reviewers document and Cover letter document, both of which have been uploaded.

---

## [Editor Report · Decision Letter 1]

11 Oct 2021

Effects of cannabis ingestion on endometriosis-associated pelvic pain and related symptoms.

PONE-D-21-19050R1

Dear Dr. Sinclair,

We’re pleased to inform you that your manuscript has been judged scientifically suitable for publication and will be formally accepted for publication once it meets all outstanding technical requirements.

Kind regards,

Diego Raimondo

Academic Editor

PLOS ONE

---

## [Editor Report · Acceptance letter]

18 Oct 2021

PONE-D-21-19050R1 

Effects of cannabis ingestion on endometriosis-associated pelvic pain and related symptoms. 

Dear Dr. Sinclair:

I'm pleased to inform you that your manuscript has been deemed suitable for publication in PLOS ONE. Congratulations! Your manuscript is now with our production department. 

Kind regards, 

on behalf of

Dr. Diego Raimondo 

Academic Editor

PLOS ONE